# Lateral Stability Control of Four-Wheel-Drive Electric Vehicle Based on Coordinated Control of Torque Distribution and ESP Differential Braking

**Liqing Chen** , **Zhiqiang Li, Juanjuan Yang and Yu Song** *

College of Engineering, Anhui Agricultural University, Hefei 230036, China; lqchen@ahau.edu.cn (L.C.);
ZhiQiangLi@ahau.edu.cn (Z.L.); yangjuanjuan@ahau.edu.cn (J.Y.)
* Correspondence: songyu@ahau.edu.cn

**Abstract:** This research focuses on four-wheel-drive electric vehicles. On the basis of the hierarchical coordinated control strategy, the coordinated control system of driving force distribution regulation and differential braking regulation was designed to increase the electric vehicles steering stability under special road working conditions. A seven-degree-of-freedom model of an electric vehicle was established in MATLAB/Simulink, and then a hierarchical coordination control model of the Electronic stability program and dynamic torque distribution control system was established. Adaptive fuzzy control was applied to ESP and, based on the neural network PID control, a torque distribution control system was designed. On the basis of the proposed coordinated control model, a performance simulation and a hardware-in-the-loop test of the control system under the typical working condition of single line shift were carried out. From the final results, it can be seen that the proposed control strategy can greatly improve the safety of the vehicle after serious side slip, increase the stability of the whole vehicle, and effectively increase the vehicle lateral stability.

**Keywords:** four-wheel-drive electric vehicle; ESP differential braking; coordination control; torque distribution; hardware-in-the-loop

## 1. Introduction

With the increasing use of green energy, electric vehicles (EVs) are becoming increasingly popular and are one of the most promising technologies in terms of transforming the transportation system [1,2]. This is because they have the advantages of energy saving and environmental protection, accurate torque control, and a simple mechanical structure [3,4]. In particular, four-wheel-drive electric vehicles have great potential in terms of improving the stability of vehicles because of their unique driving mode [5,6]. For this reason, they are becoming more and more popular with consumers. As we all know, the stability of electric vehicles in emergencies is very important. When the vehicle is in danger of instability, it is difficult for the ground to generate enough lateral force to maintain vehicle tracking and attitude adjustment [7,8]. How to enhance vehicle stability in a variety of complex environments has become a hot topic for scholars. In order to distribute the torque between the driving wheels, a lot of control strategies have been put forward.

Regarding improving the driving stability of four-wheel-drive vehicles, direct yaw moment control has a wide range of applications in vehicle driving stability control and has been widely applied [9–11]. In [12], on the basis of limited time control technology, a direct yaw moment control strategy for 4WD electric vehicles was proposed, and a torque distribution controller was constructed using limited time control technology and a nonlinear disturbance observer, which simultaneously drive the yaw angular velocity and the sideslip angle to the desired value. In order to increase the handling stability of the wheeled electric vehicle (IEV), a direct yaw moment control system was designed [13,14], which is composed of a servo-loop controller and a main loop controller. Moreover, aiming

at the four-wheel-motor independently driving electric vehicle (4MIDEV), based on the energy-saving torque distribution algorithm, a continuous steering stability controller was proposed, which not only ensures steering stability, but also improves vehicle energy efficiency. Of course, direct yaw moment control can ensure the driving stability of the vehicle. If the controller controls the vehicle after judging the vehicle to be instable, there may be some delay in the steering stability control under extreme conditions. If the vehicle instability can be predicted, it is very helpful for the timeliness of the control.

To increase the driving stability of four-wheeled independent electric vehicles, the yaw moment is determined by particle swarm optimization, adaptive control, and global optimization [15–17]. However, these studies are only applicable to simple hierarchical control methods. Ensuring the current state can effectively improve the robustness and accuracy of the system [18,19]. Lin et al. calculated the yaw moment required for dynamic control of four-wheel-drive electric vehicles through the classic sliding mode control theory [20]. Using the mathematical programming method to allocate wheel torque control with yaw torque control deviation, the energy loss of the driving system, and the slip rate constraint as penalty functions. Even if the parameters of the system are not certain, the control method is robust; therefore, it is widely used in the control for vehicle stability [21–25]. However, as a result of the high-frequency switching of the sliding mode controller, a "chattering" phenomenon occurs to a certain degree, which has a certain impact on the control accuracy of vehicle dynamics. Huang et al. studied model predictive control theory and designed a yaw moment decision controller [26]. Moreover, on the basis of the theory of tire friction ellipse, the wheel torque was distributed by minimizing the sum of the adhesion coefficients of the four tires. In the study of Nahidi et al., the advanced model predicts the optimization process of the longitudinal force and yaw moment required by the controller [27]. The low-level controller design is based on the advanced control input to better adjust the torque controller of each wheel, and the drive system distributes the required torque between the wheels.

Most of these studies are based on driving torque distribution and other joint control methods to regulate the stability of the vehicle. In addition, the electronic stabilization program (ESP) is often used to improve vehicle stability, and an ESP controller that acts on the braking system can reduce the vehicle speed and lateral acceleration to prevent a rollover. The influence of torque distribution and ESP differential braking on the additional yaw moment of automobiles was analyzed. In that study, it was proven that the differential braking and torque distribution can adjust the instability state of the vehicle steering process. Moreover, according to the fuzzy logic theory, ABS and ESP were combined to achieve stable control of the vehicle under complex braking conditions [28]. Cheng et al. used the electronic stability program (ESP) to improve tractor passability and steering stability [29]. Artificial neural networks are the second way to simulate human thinking. These are nonlinear dynamic systems that are characterized by distributed storage and parallel collaborative processing of information. The network system, composed of a large number of neurons, can achieve extremely rich and colorful behavior [30,31], In order to study and enhance the control ability of the ESP system, a complete tractor model of the steer-by-wire system and the ESP system was established. On the basis of the integrated control of braking and driving, Wang et al. established a dynamic equivalent system of vehicle stability control based on non-steady-state constraints using the control principle of artificial neural networks and ESP [32]. In particular, the BP neural network algorithm can approximate any function in theory, and its basic structure is composed of nonlinear variable elements, which have a strong nonlinear mapping ability. Therefore, we believe that using the BP neural network algorithm for torque distribution control of four-wheel-drive electric vehicles is a very interesting method [33,34].

In this study, for the purpose of controlling the electric four-wheel-drive vehicles steering stability under special road and working conditions, drive torque distribution and ESP differential braking were comprehensively utilized. The control strategy of ESP differential braking and torque distribution stratified control is put forward and a

coordination controller was designed. A hierarchical coordination control model of a seven-degree-of-freedom vehicle model, an automotive electronic stability control program, and a torque distribution control system was built with MATLAB/Simulink. Adaptive fuzzy control was adopted for ESP, and a torque distribution control system was designed through neural network PID control. A performance simulation of the control system under typical working conditions, such as a single line and step, was carried out. The results indicate that the proposed control method can guarantee the improvement of the stability and comfort of the whole vehicle and the safety of the ESP after serious sideslip.

The structure of this paper is as follows: The second section briefly presents the seven-degree-of-freedom dynamic model of the electric vehicle and the problem formulation. The design of the torque distribution controller, ESP differential braking controller, and the control strategy is shown in Section 3. In Section 4, the proposed coordinated control system is simulated and verified. Hardware-in-the-loop tests are described in Section 5 to verify the proposed control system.

## 2. Vehicle Model and Problem Formulation

First, we propose a seven-degree-of-freedom dynamic model of the vehicle in this section. Then, the tire model is proposed. Finally, the formulation of the problem is put forward.

### 2.1. DOF Vehicle Model

As shown in Figure 1, a four-wheel-drive electric vehicle dynamic model can be built into a seven-degree-of-freedom system, including longitudinal movements around the $X$ axis, lateral movements around the $Y$ axis, yaw movements around the $Z$ axis, and the rotation dynamics of the four wheels. The following simplifications were made to the vehicle:

(1) The center of the dynamic coordinate system is at the same position as the center of gravity of the vehicle;
(2) Regardless of the role of the suspension, the car is traveling in a position parallel to the ground;
(3) The effect of the steering system is neglected and the front wheel is used as input directly;
(4) The mechanical properties of the four tires are the same.

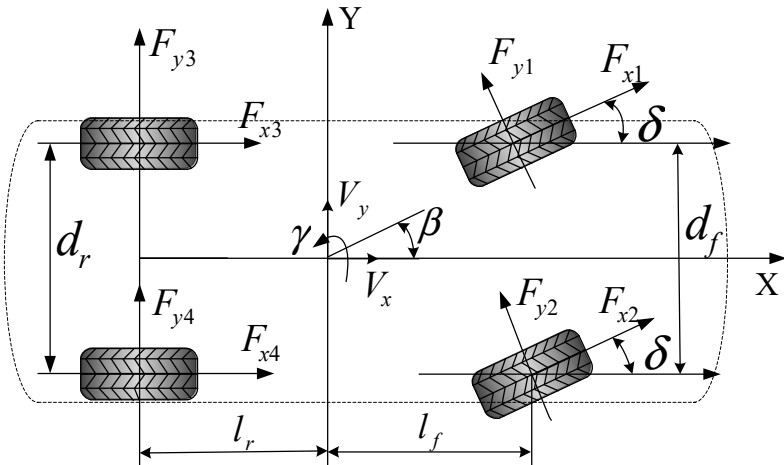

**Figure 1.** Four-wheel-drive electric vehicle model.

In Figure 1, $l_f$ represents the distance from the front axle to vehicle center of mass; $l_r$ represents the distance from the rear axle to the vehicle center of mass; $F_{xi}$ and $F_{yi}$ represent the tire longitudinal and lateral forces, respectively; $\gamma$ represents the yaw rate; $d_f$ and $d_r$ indicate the front and rear track, respectively; $\beta$ represents the electric vehicle side slip angle; $\delta$ represents the front wheel angle ($\delta \geq 0$ in the counter clockwise direction; $\delta < 0$ in

the clock direction); $V_x$ represents vehicle longitudinal speed; $V_y$ indicates the longitudinal speed of the vehicle lateral speed. The subscript $j$ indicates the left ($l$) wheel or the right ($r$) wheel, and $i$ indicates the front ($f$) wheel or the rear ($r$) wheel.

The longitudinal movement around the X axis is described as:

$$m\left(\dot{V}_x + V_y \frac{d\gamma}{dt}\right) = F_{x3} + F_{x4} + (F_{x1} + F_{x2})cos\delta - (F_{y1} + F_{y2})sin\delta$$
$$-(F_{z1}f_1 + F_{z2}f_2)cos\delta - F_{z3}f_3 - F_{z4}f_4 - \frac{C_D A}{21.15}V_x^2 \tag{1}$$

The lateral movement is modeled as

$$m\left(\dot{V}_y + V_x \frac{d\gamma}{dt}\right) = F_{y3} + F_{y4} + (F_{x1} + F_{x2})sin\delta + (F_{y1} + F_{y2})cos\delta - F_{z3}f_3$$
$$-F_{z4}f_4 - (F_{z1}f_1 + F_{z2}f_2)sin\delta \tag{2}$$

The yaw dynamics is given as

$$J_z\dot{\gamma} = \left(-\frac{d_f}{2}cos\delta + l_f sin\delta\right)F_{x1} + \left(\frac{d_f}{2}cos\delta + l_f sin\delta\right)F_{x2} + \frac{d_r}{2}(F_{x4} - F_{x3})$$
$$+\frac{d_f}{2}sin\delta(F_{y1} - F_{y2}) + l_f cos\delta(F_{y1} + F_{y2}) - l_r(F_{y3} + F_{y4}) \tag{3}$$

where $m$ refers to the vehicle weight; $g$ refers to the acceleration of gravity; $f_1, f_2, f_3, f_4$ refer to the coefficient of rolling resistance; $A$ refers to the windward area; $J_z$ is the yaw moment of inertia; and $C_D$ refers to the air drag coefficient.

The dynamics of wheel rotation are expressed as

$$J_i \cdot \dot{\omega}_i = T_{xi} - T_{bi} - T_{fi} - F_{xi} \cdot R \tag{4}$$

where $J_i$ represents the moment of inertia of the wheel; $\omega_i$ represents the angular speed of the wheel; $T_{xi}$ represents the driving torque; $T_{bi}$ represents the braking torque; $T_{fi}$ represents the rolling resistance torque; and $R$ represents the rolling radius of the tire.

The rolling resistance torque of each wheel in motor vehicle movement is expressed as follows:

$$T_{fi} = f \cdot F_{zi} \cdot R_i (i = 1,2,3,4) \tag{5}$$

$$\begin{cases} F_{z1} = \frac{m}{l}\left(\frac{g \cdot l_r}{2} - \frac{a_x \cdot h_g}{2} - \frac{a_y \cdot h_g \cdot l_r}{d_f}\right) \\ F_{z2} = \frac{m}{l}\left(\frac{g \cdot l_r}{2} - \frac{a_x \cdot h_g}{2} + \frac{a_y \cdot h_g \cdot l_r}{d_f}\right) \\ F_{z3} = \frac{m}{l}\left(\frac{g \cdot l_f}{2} - \frac{a_x \cdot h_g}{2} - \frac{a_y \cdot h_g \cdot l_f}{d_r}\right) \\ F_{z4} = \frac{m}{l}\left(\frac{g \cdot l_f}{2} - \frac{a_x \cdot h_g}{2} + \frac{a_y \cdot h_g \cdot l_f}{d_f}\right) \end{cases} \tag{6}$$

where $F_{zi}(i = 1,2,3,4)$ refers to the vertical dynamic load of each tire; the longitudinal and lateral acceleration of vehicles are $a_x$, $a_y$, respectively; $h_g$ is the height of the vehicle's center of gravity; and $l$ is the wheelbase of the vehicle.

### 2.2. Tire Model

For the purpose of producing exact tire forces, it is necessary to use the appropriate tire model. Currently, the most commonly used tire models are Dugoff models, Magic Formula, and UniTire models. Because the analysis mainly involves longitudinal and lateral adhesion, in this paper, we decided to use the Dugoff tire model [35].

The slip angle $\alpha i$ and longitudinal slip ratio $s_{xi}$ are inputs to the Dugoff tire model, and the lateral force $F_{yyi}$ and longitudinal force $F_{xxi}$ are outputs. The basic formula of Dugoff tires is

$$\begin{cases} F_{xxi} = C_{xi}\frac{S_{xi}}{1 + S_{xi}}f(s) \\ F_{yyi} = C_{yi}\frac{tan(\alpha i)}{1 + S_{xi}}f(s) \end{cases} \tag{7}$$

where $F_{xxi}$ and $F_{yyi}$ represent the tire longitudinal and lateral forces, respectively; $s_{xi}(i = 1, 2, 3, 4)$ represents the tire longitudinal slip rate; and $\alpha i$ denotes the slip angle. $C_{xi}, C_{yi}$ denote the longitudinal stiffness and lateral stiffness of tires stiffness factor, respectively; and $f(s)$ denotes the function related to dynamic parameters of a tire.

### 2.3. Problem Formulation

In the vehicle yaw moment control system, by designing a control strategy, the sideslip angle and yaw rate can reach the ideal value at the same time. In order to acquire the ideal vehicle performance, models generally follow the control strategy, that is, adjusting the steering characteristics of electric vehicles to the steering characteristics of a linear 2DOF vehicle model. The model can well reflect the basic characteristics of vehicle handling stability. Therefore, this model is selected as the reference model of vehicle control, including yaw motion and lateral motion, which can be described in the following form:

$$
\begin{aligned}
m\left(\dot{V}_y + V_x \frac{d\gamma}{dt}\right) &= \left(C_f + C_r\right)\beta + \frac{1}{V_x}\left(l_f C_f - l_r C_r\right)\gamma - C_f \delta \\
J_z \dot{\gamma} &= \left(\left(l_f C_f - l_r C_r\right)\right)\beta + \frac{1}{V_x}\left(l_f{}^2 C_f - l_r{}^2 C_r\right)\gamma - l_f C_f \delta
\end{aligned}
\tag{8}
$$

where $C_f$ represents the turning stiffness of the front wheels, and $C_r$ represents the turning stiffness of the rear wheels.

The yaw rate and sideslip angle are restricted by the conditions of the road surface, and they need to meet the following constraints:

$$
\begin{cases}
\gamma_d \leq 0.85 \cdot \frac{\mu g}{V_x} \\
\beta_d \leq \mu g\left(\frac{l_r}{V_x{}^2} + \frac{m l_f}{C_r l}\right)
\end{cases}
\tag{9}
$$

where $\mu$ represents the tire-road friction coefficient; $\gamma_d$ represents the nominal yaw rate; $\beta_d$ represents the nominal sideslip angle; and $l$ represents the distance from the rear axle to the front axle.

The purpose of this paper was to design a coordinated controller based on torque distribution and ESP differential braking, with the torque distribution strategy making the slip angle and yaw rate as close to the expected value as possible. A hierarchical control system diagram of the electronic stability program and dynamic torque distribution control system is shown in Figure 2, which includes the lower controller and the upper controller. The upper controller identifies the unstable state of the electric vehicle by the expected yaw rate and the actual yaw speed parameters of the receiving system model, and determines which control strategy to use. The lower layer controller includes the ESP differential brake fuzzy controller and the dynamic torque system neural network PID controller's two subcontrollers. The driving force and braking force of the vehicle are allocated according to the upper-level decision, so as to ensure the electric vehicle is stable.

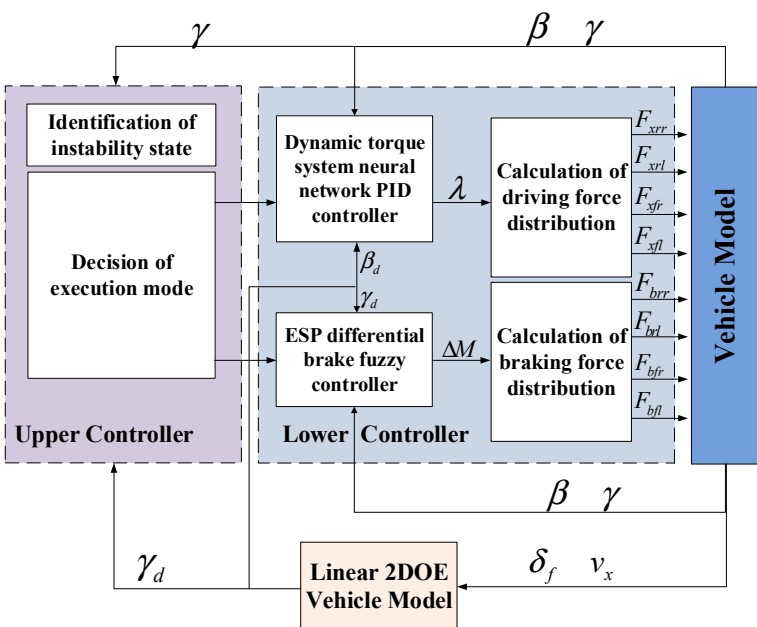

**Figure 2.** Hierarchical control system.

## 3. Control System Design

The traditional ESP control is the braking control at the cost of power consumption. Inside a certain range, the vehicle attitude can be stabilized by power distribution, but beyond this range, ESP intervention is needed. It is our expectation to establish a coordination mechanism between the ESP system and the power distribution system, and to control the vehicle attitude through power distribution between axles and wheels in a certain range, so as to avoid ESP intervention to the maximum extent. This is because, for electric vehicles, adjusting the vehicle attitude by braking uses lots of power. Therefore, in this section, a dynamic torque system neural network PID controller is proposed. Then, an ESP differential brake fuzzy controller is proposed thereafter. Finally, an upper controller based on two party coordinated controls is constructed, and the yaw moment is allocated to four wheels based on the coordinated control strategy.

### 3.1. Dynamic Torque System Neural Network PID Controller Design

3.1.1. Torque Distribution Model

Additional yaw moments are urgently needed to correct the steering attitude and improve the lateral stability of automobiles when steering instability occurs. The transfer of driving forces between wheels and axles causes changes in the longitudinal and lateral forces of the tires, which affect the yaw characteristics of the vehicle under this steering [36,37]. Previous studies discuss the influence of driving force transfer between the axles and the wheels on the extra yaw moment of the vehicle. It is known that the driving force transfer between the axles has less influence on the state of the vehicle, under the ratio of driving force distribution between different axles. Moreover, the sideslip angle and yaw rate change are very limited, but the vehicle state can be changed obviously by the distribution of driving force between wheels.

The electric transfer case designed in this paper was designed with a multiplate clutch. Its working principle is to adjust the compression degree of the multiplate clutch by turning the motor, so as to change the power distribution. It has nonlinear characteristics. Therefore, on the basis of the actual situation of the torque distribution system, we assume that the torque distribution coefficient of the front axle of the car is $\alpha$, and the torque distribution coefficients between the right and the left wheels of the front axle and rear axle are $\lambda_1, \lambda_2$

respectively. The torque transmitted by the transmission system to each wheel can be described as

$$\begin{cases} T_{x1} = \alpha T_e i_{gi} i_0 \eta \lambda_1 \\ T_{x2} = \alpha T_e i_{gi} i_0 \eta (1 - \lambda_1) \\ T_{x3} = (1 - \alpha) T_e i_{gi} i_0 \eta \lambda_2 \\ T_{x4} = (1 - \alpha) T_e i_{gi} i_0 \eta (1 - \lambda_2) \end{cases} \tag{10}$$

where $T_{X1}, T_{X2}, T_{X3}, T_{X4}$ denote the driving torque of the four driving wheels and the braking torque applied to each wheel, respectively; $i_{gi}$ refers to the transmission ratio; $i_0$ refers to the drive ratio of main decelerator; and $\eta$ refers to the transmission efficiency.

### 3.1.2. Design of PID Controller Based on BP Neural Network

The BP neural network has a strong nonlinear solution and adaptive ability, so it can handle the nonlinearity of the entire vehicle [38,39] in order to adjust the torque distribution coefficient of electric vehicles, based on the BP neural network designed PID controller. The controller is mainly composed of the following two parts:

(1) The classic PID controller: a closed cycle model was formed with the whole system to achieve the purpose of controlling feedback, and an incremental PID control algorithm was used, the output is

$$u(k) = u(k-1) + K_p[e(k) - e(k-1)] + K_i e(k) + K_d [e(k) - 2e(k-1) + e(k-2)] \tag{11}$$

(2) The BP neural network: it can quickly adjust the weight coefficients between neurons according to the state of the system, so as to obtain the optimum PID control parameters. The network includes an input layer, an output layer, and a hidden layer; it is determined as a 4-5-3 structure according to the studied system. The required yaw rate, the actual yaw rate, and the error between them are selected as the network inputs (when the vehicle sideslip angle is greater than the stable boundary, which is calculated on the basis of the road and speed, the input is switched to the corresponding vehicle sideslip angle), and, for the purpose of stabilize the network, the constant term 1 is added.

The following form can be used to represent the input of the BP neural network:

$$X(j) = [r(k), y(k), e(k), 1] \tag{12}$$

Then, the input layer node output is

$$O_{(j)}^1 = X(j) \tag{13}$$

The input and output of hidden neurons layer can be expressed as

$$\begin{cases} net_i^2(t) = \sum_{j=0}^{m} w_{ij}^{(2)} O_j^{(1)} \quad i = 1, 2, \cdots\cdots, q \\ O_i^{(2)}(t) = g\left(net_i^2(t)\right) \end{cases} \tag{14}$$

where $W_{ij}{}^2$ denotes the weighting coefficient of the input layer to the hidden layer; $g(\cdot)$ denotes the transfer function; and the Sigmoid function is shown below:

$$g(x) = \frac{1}{1 + e^{-x}} \tag{15}$$

The total input and output of neurons in the output layer are

$$\begin{cases} net_k^3(t) = \sum_{i=0}^{q} W_{ik}^{(3)} O_i^{(2)}(t) \quad k = 1, 2, 3 \\ O_k^{(3)}(t) = f\left(net_k^3(t)\right) \end{cases} \tag{16}$$

where $W_{ik}^{(3)}$ denotes the weight coefficient between the output layer and the hidden layer; and $f(\cdot)$ denotes the output transfer function.

When there is a difference between the expected output and the actual output, the back propagation of the error is adjusted to make the output value close to the expected value, and the target function is selected as

$$E = \frac{1}{2}\sum_{k=1}^{m}(d_k - o_k)^2 = \frac{1}{2}\sum_{k=1}^{m}e_k^{\,2} \tag{17}$$

In order to make the network converge, we chose the gradient descent method to regulate the weights of the neural network in real time. Figure 3 is the structure of the torque distribution control system. The output of the PID controller is the distribution coefficient of the torque between the wheels. The control variable selection module switches the input state of the neural network by judging the vehicle sideslip angle.

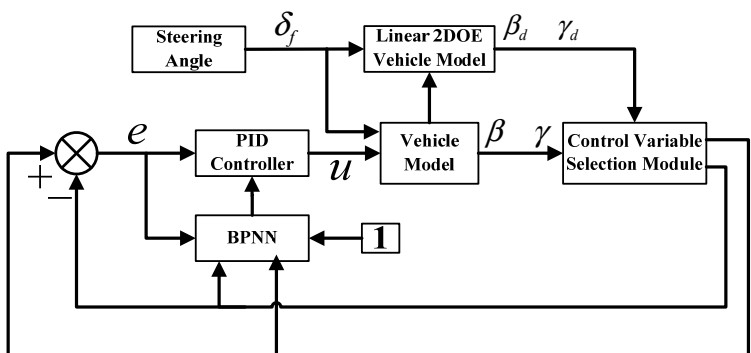

**Figure 3.** Torque distribution control system.

Under the steering condition, the control system intelligently distributes torque by identifying the vehicle sideslip and yaw characteristics of the whole vehicle, so that the vehicle maintains good handling and driving safety. When the vehicle's sideslip angle is small, the yaw rate is used to control the whole vehicle state. Otherwise, the vehicle sideslip angle is regarded as the main control object.

### 3.2. ESP Differential Brake Fuzzy Controller Design

3.2.1. ESP Differential Braking System Model

The brake torque of disc brake can be expressed as

$$T_b = A \cdot n \cdot \mu \cdot R_3 \cdot \eta_b \cdot P_b \tag{18}$$

where $A$ denotes the contact area between brake caliper and brake disc; $n$ is the number of single wheel brake calipers; $\mu$ denotes the friction coefficient during braking; $R_s$ is the effective braking radius; and $\eta_b$ is the brake efficiency.

When the automotive steering is unstable, the ESP can produce the corresponding additional yaw moment by applying the active braking force to a single or various side wheels, thereby adjusting the posture of the vehicle body and maintaining the vehicle body stability. The extreme limiting power obtained by a single wheel of a vehicle has a very important relationship with the wheel adhesion coefficient and the dynamic load of the tire, and satisfies the following formula:

$$F_{bi} \leq \mu \cdot F_{zi} \tag{19}$$

Taking the unilateral brake as an example, the additional yaw moment is only produced by the unilateral differential brake and has nothing to do with the lateral force.

Suppose that the vehicle ESP exerts an active braking force on the outer wheel during left steering, the additional yaw moment of the vehicle is as follows:

$$\Delta M_b = \frac{1}{2}F_{b1}\cdot cos\delta_f\cdot d_f + F_{b1}\cdot sin\delta_f\cdot l_f + \frac{1}{2}F_{b3}\cdot d_r \approx \frac{1}{2}F_{b1}\cdot d_f + \frac{1}{2}F_{b3}\cdot d_r \tag{20}$$

According to Equation (20), the additional yaw moment generated by unilateral differential braking is determined by the braking force exerted on the side wheel. Under the same road and driving conditions, the torque of the wheel is certain, and the wheel braking force is determined by the cylinder pressure of the brake wheel and it can be regulated in real time within the limit range. Therefore, the control amplitude of the additional yaw moment of the differential brake is greater than the control of the torque distribution.

### 3.2.2. Fuzzy Controller Design

The fuzzy control system can handle the uncertain relationship of various complex systems [40], so in order to track the expected state of electric vehicle, a fuzzy controller was designed. The difference between the expected and actual yaw velocity and centroid angle are taken as two vague input variables, and the modifier of the yaw moment is the output variable. The input and output are all triangular membership functions. The fuzzy rules are shown in Table 1.

**Table 1.** Fuzzy control rules.

| $e(\gamma)$ | $e(\beta)$ | | | | | | | |
|---|---|---|---|---|---|---|---|---|
| | **NB** | **NM** | **NS** | **ZE** | **PS** | **PM** | **PB** | **NB** |
| NB | PB | PB | PB | PM | PB | PB | PB | PB |
| NM | PB | PB | PB | PM | PB | PB | PB | PB |
| NS | PS | PS | NS | PS | PM | PM | PB | PS |
| ZE | ZE | ZE | NS | ZE | PS | ZE | ZE | ZE |
| PS | NB | NM | NM | NS | PS | NS | NS | NB |
| PM | NB | NB | NB | NM | NB | NB | NB | NB |
| PB | NB | NB | NB | NM | NB | NB | NB | NB |

The application of the differential brake to adjust the yaw moment of the whole vehicle has been widely studied. Taking the unilateral wheel brake as an example, the wheel braking force and the additional yaw moment relationship is as follows:

$$\begin{cases} T_{bi} = \frac{F_{zi}}{F_{z(i+2)+F_{zi}}} \cdot \frac{4\cdot\Delta M}{d_f + d_r} R \\ T_{b(i+2)} = \frac{F_{z(i+2)}}{F_{z(i+2)+F_{zi}}} \cdot \frac{4\cdot\Delta M}{d_f + d_r} R \end{cases} \tag{21}$$

where $i$ is 1 or 3; if the vehicle is understeering, apply braking force to the outer wheels; if oversteering, apply braking force to the inner wheels.

### 3.3. Upper Controller Design

As a result of the dynamic analysis of the power distribution model and the differential brake model, the dynamic torque system neural network PID controller and ESP differential braking fuzzy controller have the control effect on the yaw moment of the whole vehicle, and the control range of the additional yaw moment of the entire vehicle by the differential brake is larger than that of the driving force distribution control.

After the upper controller receives the signal, the judgment variable is set as follows:

$$c = |\gamma_d - \gamma| \tag{22}$$

where $\gamma$ is the actual yaw rate; and $\gamma_d$ denotes the expected yaw rate.

It is known from the simulation that, when $c < 0.02$, the electric vehicles understeer or oversteer.

The torque of the right and left wheels can be distributed in real time by the dynamic torque system neural network PID controller, so that the driver's steering intention is effectively tracked while the power of the electric vehicle is guaranteed. When $c \geq 0.02$, the electric vehicle is in a status of serious instability, and there is a need for the ESP differential braking fuzzy control to intervene to ensure vehicle safety.

## 4. Control System Simulation Validation

As to verify the effectiveness of the proposed control system and the controller, the model under general operating conditions was simulated using MATLAB/Simulink software. Table 2 shows the parameters of the electric vehicle model, and this selected electric vehicle is a traditional driving structure. The driving force and torque are output by the central motor and distributed to the rear and front wheels through the distributor and differential brake control. The motor type is TZ200XSJH7. The torque output characteristics of the motor reducer assembly were measured in experiments as shown in Figure 4.

**Table 2.** Electric vehicle parameters.

| Definition | Symbol | Value (Unit) |
| --- | --- | --- |
| Vehicle weight | $m$ | 1760 (kg) |
| Gravitational acceleration | $g$ | 9.8 (m/s$^2$) |
| Yaw moment of the inertia | $l_z$ | 3100 (kg·m$^2$) |
| Spin inertia of the wheel | $l_w$ | 2.1 (kg·m$^2$) |
| Wheelbase | $l$ | 20,611(m) |
| Distance from c front axle to entroid | $l_f$ | 1.219 (m) |
| Distance from rear axle to centroid | $l_r$ | 1.392 (m) |
| Distance from ground to centroid | $h_g$ | 0.52 (m) |
| Front track | $d_f$ | 1.52 (m) |
| Rear track | $d_r$ | 1.52 (m) |
| Rolling radius of the tire | $R$ | 0.304 (m) |
| Front wheel turning stiffness | $C_f$ | 41,310 |
| Rear wheel turning stiffness | $C_r$ | 45,218 |
| Wheel longitudinal stiffness | $C$ | 50,000 |
| Rolling resistance coefficient | $f$ | 0.018 |
| Windward area | $A$ | 2.88 |

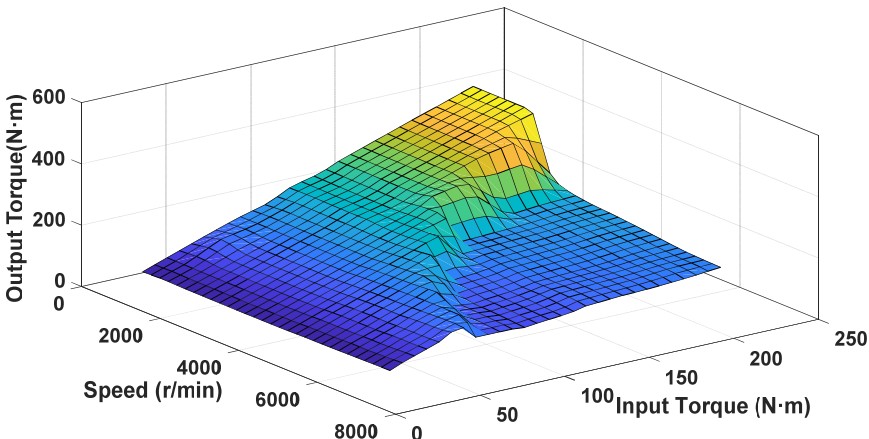

**Figure 4.** The torque output characteristics of motor reducer assembly.

The single lane-change simulation test for electric vehicles was carried out on both low adhesion and high adhesion coefficient pavements.

### 4.1. Simulation with Low Adhesion Coefficient Pavement

When the steering wheel is turned to a certain angle, vehicles with low adhesion may create the risk of yaw instability on the road. The effect of the control system on vehicle stability was verified by selecting one-way conditions on low-adhesion roads. The simulation conditions were set as follows: the driving speed was 60 km/h; the road adhesion coefficient was 0.3; and single lane-change lane running under front wheel angle 0.05 rad. The response curve of the yaw rate, side angle, and longitudinal speed of the noncontrol, torque distribution control, and differential braking control are given in Figure 5a–c.

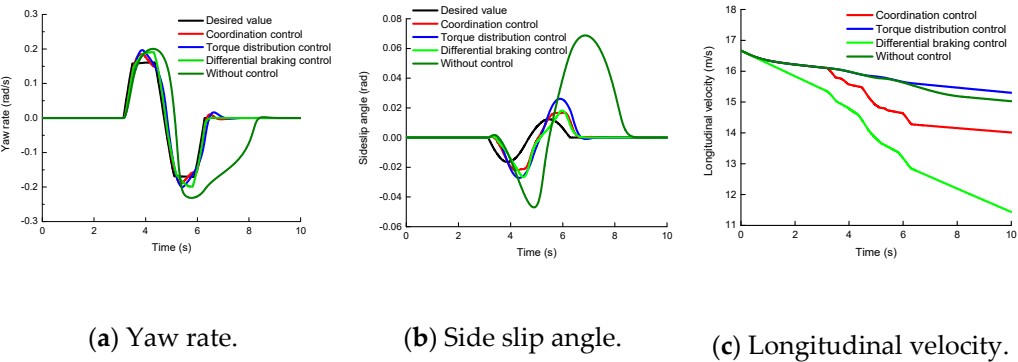

(**a**) Yaw rate.    (**b**) Side slip angle.    (**c**) Longitudinal velocity.

**Figure 5.** Single lane simulation test on low adhesion pavement.

According to Figure 5, the torque distribution control and differential braking control can adjust the whole vehicle state well as compared with the vehicle without the control. The yaw rate can be closely followed by the expected value, and the side angles of the centroids were also reduced to 0.026 rad and 0.018 rad from 0.069 rad without control, which shows that the torque distribution control and differential braking control can play a certain role in the control of the whole vehicle state.

### 4.2. Simulation with High Adhesion Coefficient Pavement

The initial speed of the car was 80 km/h, and the single line change path was initiated after some seconds at steady speed. The adhesion coefficient of the road surface was 0.6 and the front-wheel angle was 0.12 rad. The simulation results are given in Figure 6a–c.

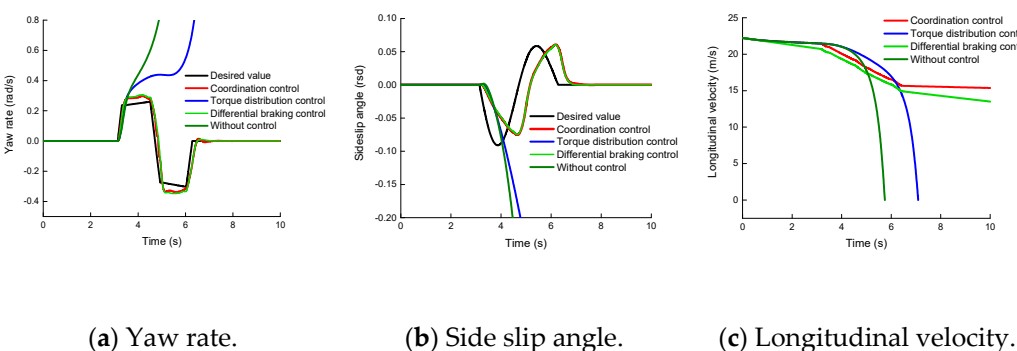

(**a**) Yaw rate.    (**b**) Side slip angle.    (**c**) Longitudinal velocity.

**Figure 6.** Single lane simulation test on high adhesion pavement.

In Figure 6, it is shown that when the vehicle is seriously unstable, the vehicle state cannot be controlled within a safe range through the torque distribution mode, and the sideslip angle and yaw rate of the vehicle range widely; thus, it is impossible to effectively track the expected value.

The differential braking control was able to stabilize the state of the whole vehicle near the expected value, because the peak of the yaw rate was only slightly higher than the expected value of 0.046 rad/s, and the side angle of the centroid was controlled within

0.060 rad. However, as shown in Figure 6c, it is obvious that the speed loss under the control of differential braking was larger, indicating that the use of the differential brake control in normal driving does not offer excellent operating performance. The proposed coordination control system emphasizes the advantages of the two, ensures the stability and maneuverability of the body under general sideslip, and ensures the safety of the vehicle when the sideslip is serious. As can be seen from Figure 6, the yaw rate was only 0.037 rad/s higher than the expected value under the coordinated control system, the peak of the centroid side angle was 0.061 rad, and the speed loss was smaller than with the differential brake control, which proves that the vehicle runs more stably under the coordinated control.

## 5. Verification of Hardware-In-The-Loop Test

### 5.1. Development of Hardware-In-The-Loop Test Platform

On the basis of the NI PXI real-time processor, a hardware-in-the-loop test consisting of a real controller and a virtual controlled object was performed to further verify the effectiveness of the proposed vehicle state control system. The hardware platform of the control system mainly included a NI real-time processor, a personal computer (PC), a VG 440C A gyroscope, a router, a steering parameter tester, a TCP/IP bus, and related sensors. As the host computer, a PC was mainly used to run the hardware-in-the-loop real-time measurement and control system based on NI Veristand and LabVIEW. Moreover, the PC acted as the human–computer interaction interface to monitor and control the whole hardware platform in real time and complete the mapping between the 7-DOF vehicle model and the controller. The PXI-8102 real-time controller was equipped with LabVIEW Real-Time operating system, which was used as a slave computer to download and run the controller DLL files, which were compiled by the MATLAB/RTW (Real-time workshop) module; the TCP/IP bus was used as a bridge to connect the PXI-8102 real-time processor with the computer to complete the network architecture. The system deployed the transformed MATLAB/Simulink model to the PXI real-time processor of the lower computer, and carried out the test management of the whole platform, such as switch control, parameter adjustment, state excitation, and data monitoring.

### 5.2. Hardware-In-The-Loop Test

The single lane change field test of vehicles was conducted on a dry cement road according to JASO-C-707 and ISO/FDIS3888-2 standards. Because it is not easy to directly measure the front wheel angle, the steering wheel angle was selected as the data acquisition object, and the collected steering wheel angle data were processed and converted to the front wheel angle of the whole vehicle model as the input of the hardware-in-the-loop test. Figure 7 shows the real vehicle test data acquisition system.

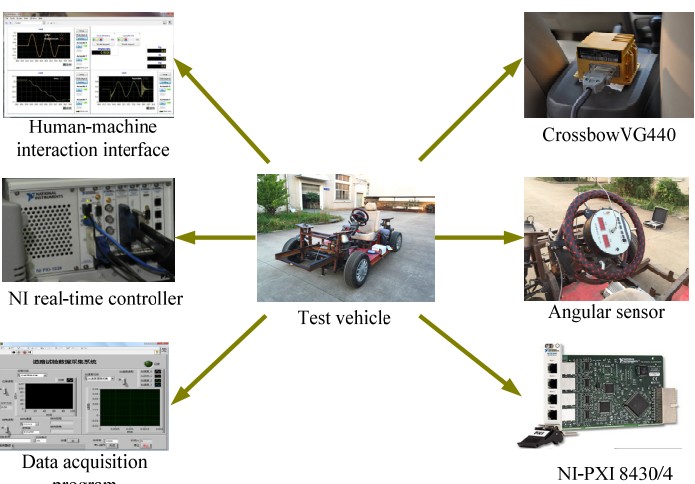

**Figure 7.** The real vehicle test system.

### 5.3. Test Results and Analysis

The gyroscope was installed in the vehicle to collect the lateral acceleration and yaw rate of the vehicle in real time. The vertical and horizontal speeds were measured with GPS speed sensors. The angular velocity of the four wheels was collected by the speed sensor of the motor in the wheel. The steering wheel angle sensor was used to measure the steering wheel angle. The test vehicle speed was 50 km/h ($\pm2$ km/h).

Figure 8 shows the front wheel angle data collected under the single lane-change driving condition, which was used as the input of the hardware-in-the-loop signal for the test simulation. In order to make the hardware-in-the-loop simulation results more clearly reflect the role of the control system, the adhesion coefficient of the simulated pavement was set to 0.3. The single lane-change test results are shown in Figure 9.

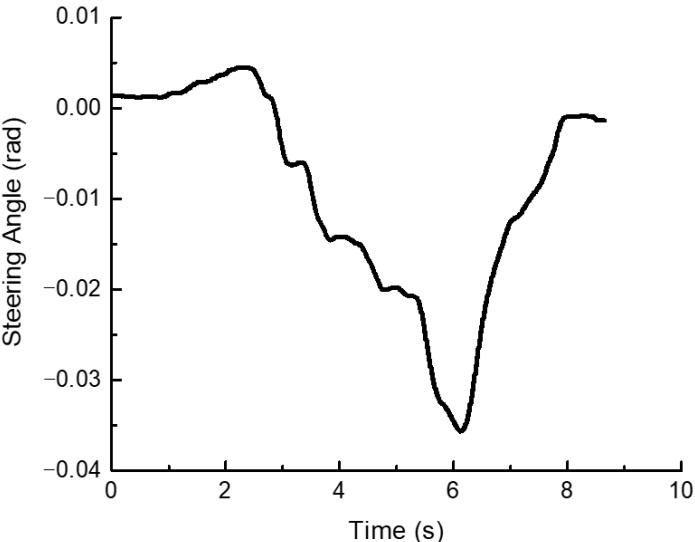

**Figure 8.** Steering wheel angle input.

The test results show that, in order to follow the target path, the front wheel angle of the vehicle exhibited irregular wavy change during continuous steering. From the Figure 8, the peak at the turn occurred during the lane change, and the driving state of the vehicle was poorest. By effectively distributing the driving torque of each wheel, the vehicle state was greatly improved. As shown in Figure 9a,b, when the vehicle was not under control, the vehicle demonstrated steering instability when changing lanes. In contrast, the vehicle state was obviously improved by the coordinated control. The designed control system was able to accurately track the desired yaw angular velocity and significantly suppress the sideslip angle, and restore the yaw angular velocity and the sideslip angle to the desired value under the instability state, so that the vehicle attitude could be corrected and the vehicle could continue driving. As compared with the without-control state, the peak yaw rate and sideslip angle were reduced by 0.043 rad/s and 0.030 rad, respectively, under the state of coordinated control. Furthermore, their magnitude was reduced by up to 21.6% and 87.4%, respectively, and the state response lag was improved to some extent. The comparison shows that the control system can speed up the dynamic response of the vehicle to the driver's intention, and thus obtain better maneuverability. The objective of this paper was to verify the feasibility of the control strategy. In order to ensure the safety and stability of the experiment, the author adopted the traditional method of combining torque vectoring of an EV with brake-based ESP. However, the method of combining the motor and braking is also worth studying. We will carry out the corresponding research in future work. Moreover, in the future, we will carry out more performance tests under different working conditions to continuously improve the performance of the controller.

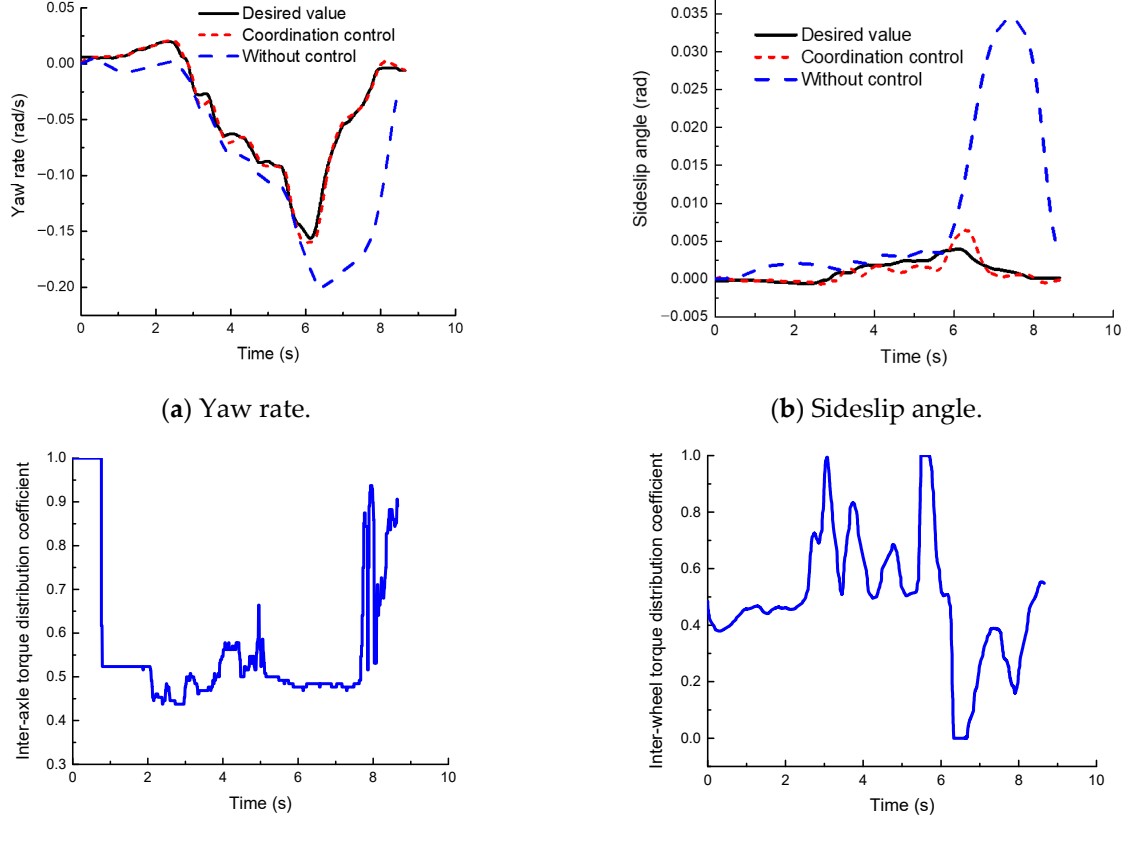

(**a**) Yaw rate.

(**b**) Sideslip angle.

(**c**) Inter-axle torque distribution coefficient.

(**d**) Inter-wheel torque distribution coefficient.

**Figure 9.** Single lane-change simulation test result.

**Author Contributions:** Conceptualization, L.C.; methodology, Z.L.; software, Z.L.; validation, J.Y.; formal analysis, J.Y.; investigation, J.Y.; resources, Z.L.; data curation, Z.L.; writing—original draft preparation, L.C.; writing—review and editing, L.C. and Y.S.; visualization, L.C.; supervision, Z.L.; project administration, Z.L.; funding acquisition, L.C. All authors have read and agreed to the published version of the manuscript.

**Funding:** This study was funded by National Natural Science Foundation of China (Grant No. 51305004) and Anhui Natural Science Foundation Project (Grant No. 1908085ME174).

**Institutional Review Board Statement:** Not applicable.

**Informed Consent Statement:** Not applicable.

**Data Availability Statement:** The data used in this study is self-test and self-collection. Because the control method designed in this paper is still being further improved, Data cannot be shared at present.

**Conflicts of Interest:** The authors declare no conflict of interest.

## Abbreviations

The following abbreviations are used in this manuscript:

| | |
|---|---|
| BP | Back Propagation |
| EVs | Electric Vehicles |
| 4WD | Four wheel drive |
| IEV | The Wheeled Electric Vehicle |
| 4MIDEV | Four-Wheel Motor Independently Driving Electric Vehicle |
| ESP | Electronic Stabilization Program |
| ABS | Anti-Lock Braking System |
| PID | Proportional Integral Differential |

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
