# Peer review of "Lateral Stability Control of Four-Wheel-Drive Electric Vehicle Based on Coordinated Control of Torque Distribution and ESP Differential Braking"

_actuators, doi:10.3390/act10060135_

Round 1
Reviewer 1 Report
The authors propose a coordinated control allocation approach for four wheel drive electric vehicle. It uses control switching between torque distribution and ESP differential braking to improve the steering stability under special road working conditions. The proposed control strategy can greatly improve the vehicle lateral stability. The topic is quite interesting, but some issues also exist shown as follows.
- The Introduction of this work could be further clarified, such as adding some latest literature reviews.
- The reference is suggested for the establishment of the relevant equation, such as Formula 1.
- The symbols appearing in the text need further proofreading, and each symbol should be annotated.
- Some future work could be included in the Conclusion section.
Author Response
Dear peer reviewers and editors
Hello! Thank you very much for your professional and wise comments on this article. According to the experts' questions and opinions, the author gives a detailed explanation in the form of one question and one answer, and marks the manuscript with red font in the corresponding position, as follows:
QUESTION1:The Introduction of this work could be further clarified, such as adding some latest literature reviews.
RE: Thank you very much for your valuable comments. The author has revised the introduction, added the latest research status, and marked the revised part with red font in the introduction.
QUESTION2:The reference is suggested for the establishment of the relevant equation, such as Formula 1.
RE: Thanks to the reviewers for their valuable comments. The author proofread the formula used in this paper carefully, for places without references, corresponding references are cited, and marked the corresponding position in the paper with red font.
QUESTION3:The symbols appearing in the text need further proofreading, and each symbol should be annotated.
RE: Thank the reviewers for their valuable comments. The author proofread the symbols used in this paper carefully, and annotated the parts without notes, and marked the corresponding positions with red font.
QUESTION4:Some future work could be included in the Conclusion section.
RE: Thanks to the reviewers for their valuable comments. The author added the future work plan for the conclusion and displayed the revised contents in red font.
Thank you very much for your valuable opinions. Please review the revised article!

Reviewer 2 Report
The paper describes a supervisor control strategy for torque vectoring and ESP for an electric vehicle with supposedly individual wheel drives, although that is not clear from the submission.
Unfortunately, the presentation of the paper is poor, so it is hard to understand what has been achieved. Wide parts of the paper reproduce textbook level background, without actually explaining how this is implemented.
There are several notable flaws in the paper. The combination of torque vectoring of an EV with brake based ESP seems a bit weird, since ESP has been around for conventional vehicles for a long time, and a rather conventional approach is taken here, while an EV would offer more interesting approaches by combining motors and brakes.
The magic formula tyre model is a bit too simple for ESP, where a more sophisticated model with maximum traction at around 20% slip is used.
The torque vectoring control is also questionable. The conventional approach is to apply a differential torque, which is close to linear. But this paper uses a torque distribution factor, which makes the model unnecessarily non-linear. It seems the same control structure is used for yaw rate and slip angle, which requires a better motivation.
A neural network control is designed, but no details are given about the neural network or the training objective. This would make it very hard to replicate the result.
The validation on the hardware in the loop rig is not entirely convincing, because it does of course crucially depend on the model. If this model is the same as the one used for designing the controller, the test will obviously succeed, but this has little predictive power for a real application.
The publication of more implementation detail is desperately needed before the quality of the paper can be adequately judged.
Author Response
Dear peer reviewers and editors
Hello! Thank you very much for your professional and wise comments on this article. This research focuses on four-wheel-drive electric vehicles. Based on the hierarchical coordinated control strategy, the coordinated control system of driving force distribution regulation and differential braking regulation is designed, to increase the electric vehicles steering stability under special road working conditions. The following is a question and answer way to reply to your comments, and highlight them in red font in the corresponding position of the paper.
QUESTION1:The combination of torque vectoring of an EV with brake based ESP seems a bit weird, since ESP has been around for conventional vehicles for a long time, and a rather conventional approach is taken here, while an EV would offer more interesting approaches by combining motors and brakes.
RE: Thank the reviewers for their valuable opinions. The electric vehicle discussed in this paper is not an existing electric vehicle driven by hub motor. Its structure is mainly a four-wheel drive vehicle driven by an electric motor, which is transmitted to the wheels through the transmission, electronic transfer case and drive axle, which is equivalent to an electric four-wheel drive off-road vehicle. Therefore, the combination of power distribution and ESP is a joint control problem based on electric drive. The purpose of power distribution is to control the vehicle attitude through power distribution between axles and wheels, while the control of ESP is the braking control at the expense of power; Through the analysis, it is found that the safety of the whole vehicle can be ensured by power distribution within a certain range, and ESP intervention is needed to control the vehicle beyond the range. Therefore, the main purpose of this paper is not to discuss the power distribution and ESP control, but mainly to discuss the coordination mechanism of the two systems. This is also the main innovation of this paper. Before the author may not express clearly in the text, now the whole text has been modified, the modified part is marked in red font in the text.
QUESTION2:The paper describes a supervisor control strategy for torque vectoring and ESP for an electric vehicle with supposedly individual wheel drives, although that is not clear from the submission.
RE: Thank the reviewers for their valuable comments. This paper uses the Dugoff tire model instead of the magic model. I hope the reviewers can check the papers, Dugoff tire model mainly involves the study of longitudinal and transverse adhesion characteristics, which can meet the requirements of this paper.
QUESTION3:The torque vectoring control is also questionable. The conventional approach is to apply a differential torque, which is close to linear. But this paper uses a torque distribution factor, which makes the model unnecessarily non-linear. It seems the same control structure is used for yaw rate and slip angle, which requires a better motivation.
RE: Many thanks to the reviewers for their valuable opinions. As mentioned earlier in this paper, the electric vehicle discussed in this paper is not an existing electric vehicle driven by a wheel motor. Its structure is mainly a four-wheel drive vehicle driven by a single motor, which is transmitted to the wheels through the transmission, electronic transfer case and drive axle. The electric transfer case used in this vehicle is an electric transfer case with multi plate clutch, which is widely used in off-road vehicles. Its working principle is to adjust the compression degree of multi plate clutch through motor rotation, so as to change the power distribution. It has nonlinear characteristics, and the motor control is the core, In this paper, the torque distribution system is also set based on the actual situation. For this part of the content, the author has added in the article. Experts are invited to review.
QUESTION4:A neural network control is designed, but no details are given about the neural network or the training objective. This would make it very hard to repli cate the result.
RE: Many thanks to the reviewers for their valuable opinions. The details of the control have been added in section 3.1.2 and marked in red
QUESTION5:The validation on the hardware in the loop rig is not entirely convincing, because it does of course crucially depend on the model. If this model is the same as the one used for designing the controller, the test will obviously succeed, but this has little predictive power for a real application.
RE: Thank the reviewers for their valuable opinions. The hardware in the loop test in this paper is designed according to jaso-c-707 and ISO / fdis3888-2 standards, and the actual data of the test vehicle is collected; On this basis, the results show that it is feasible. Of course, as the reviewers said, it is the most convincing to carry out the real vehicle control test, which is also the work that the author is currently carrying out.
Thank you very much for your valuable opinions. Please review the revised article!

Reviewer 3 Report
In this paper, the electric four-wheel drive vehicles steering stability is considered under special roads and working conditions, drive torque distribution. ESP differential braking and torque distribution are considered, and a coordination controller is designed. A hierarchical coordination control model is built through MATLAB/Simulink. Adaptive fuzzy control is adopted for ESP, and a torque distribution control system is designed through neural network PID control. The numerical simulation results are given.
My main concern is the paper novelty. Even though some papers are reviewed in the introduction, it is not clear what the contributions of this work are. In fact, considering the given introduction, the paper research direction is not motivated. I recommend considering the latest papers published and organizing the introduction such that it conveys and highlights the paper contribution and novelty. To do so, some comparison to other approaches is required.
There are lots of grammatical mistakes and typos. Please improve the paper written English, especially in the introduction.
Author Response
Dear peer reviewers and editors
Hello! Thank you very much for your professional and wise comments on this article. According to the experts' questions and opinions, the author gives a detailed explanation in the form of one question and one answer, and marks the manuscript with red font in the corresponding position, as follows:
QUESTIONS: In this paper, the electric four-wheel drive vehicles steering stability is considered under special roads and working conditions, drive torque distribution. ESP differential braking and torque distribution are considered, and a coordination controller is designed. A hierarchical coordination control model is built through MATLAB/Simulink. Adaptive fuzzy control is adopted for ESP, and a torque distribution control system is designed through neural network PID control. The numerical simulation results are given. My main concern is the paper novelty. Even though some papers are reviewed in the introduction, it is not clear what the contributions of this work are. In fact, considering the given introduction, the paper research direction is not motivated. I recommend considering the latest papers published and organizing the introduction such that it conveys and highlights the paper contribution and novelty. To do so, some comparison to other approaches is required. There are lots of grammatical mistakes and typos. Please improve the paper written English, especially in the introduction.
RE: Thank you very much for your valuable comments. In order to highlight the novelty of this paper, the author cites the latest papers in the introduction and makes some comparisons with other methods; In view of the grammar and spelling problems of the manuscript, we have also carefully checked and corrected the mistakes, marking them in red font.
Thank you very much for your valuable opinions. Please review the revised article!

Round 2
Reviewer 2 Report
The paper has been extensively edited based on the feedback from the first review round. I feel that all critical comments have been addressed, and the paper is much improved.
I am still concerned about the validation methodology, because standardised or not, it does not demonstrate any significant robustness of the approach. HIL rigs are usually used to demonstrate the correctness of the implementation of the control algorithm, not the correctness of the control algorithm itself.
Overall, this is probably as good as this paper will get, and I think it just about qualifies for publication. I would still recommend that the controller implementation is published, so that it can be precisely replicated by other researchers.
Reviewer 3 Report
In this new version, the authors have carefully addressed my comments on the previous version. Therefore, I believe this paper can be considered for publication.